# Fluoroquinolone Versus Nonfluoroquinolone Treatment of Bloodstream Infections Caused by Chromosomally Mediated AmpC-Producing Enterobacteriaceae

**DOI:** 10.3390/antibiotics9060331

**Published:** 2020-06-17

**Authors:** Sarah Grace Gunter, Katie E. Barber, Jamie L. Wagner, Kayla R. Stover

**Affiliations:** 1Department of Pharmacy, University of Mississippi Medical Center, Jackson, MS 39216, USA; sarah.gunter@grandviewhealth.com; 2Department of Pharmacy, Grandview Medical Center, Birmingham, AL 35243, USA; 3Department of Pharmacy Practice, University of Mississippi School of Pharmacy, Jackson, MS 39216, USA; kbarber@umc.edu (K.E.B.); jwagner@umc.edu (J.L.W.); 4Division of Infectious Diseases, University of Mississippi Medical Center, Jackson, MS 39216, USA

**Keywords:** AmpC, beta-lactamases, fluoroquinolones, beta-lactams, bacteremia

## Abstract

Objectives: Chromosomally mediated AmpC-producing Enterobacteriaceae (CAE) display high susceptibility to fluoroquinolones; minimal clinical data exist supporting comparative clinical outcomes. The objective of this study was to compare treatment outcomes between fluoroquinolone and nonfluoroquinolone definitive therapy of bloodstream infections caused by CAE. Methods: This retrospective cohort assessed adult patients with positive blood cultures for CAE that received inpatient treatment for ≥48 h. The primary outcome was difference in clinical failure between patients who received fluoroquinolone (FQ) versus non-FQ treatment. Secondary endpoints included microbiological cure, infection-related length of stay, 90-day readmission, and all-cause inpatient mortality. Results: 56 patients were included in the study (31 (55%) received a FQ as definitive therapy; 25 (45%) received non-FQ). All non-FQ patients received a beta-lactam (BL). Clinical failure occurred in 10 (18%) patients, with 4 (13%) in the FQ group and 6 (24%) in the BL group (*p* = 0.315). Microbiological cure occurred in 55 (98%) patients. Median infection-related length of stay was 10 (6–20) days, with a significantly longer stay occurring in the BL group (*p* = 0.002). There was no statistical difference in 90-day readmissions between groups (7% FQ vs. 17% BL; *p* = 0.387); one patient expired. Conclusion: These results suggest that fluoroquinolones do not adversely impact clinical outcomes in patients with CAE. When alternatives to beta-lactam therapy are needed, fluoroquinolones may provide an effective option.

## 1. Introduction

Gram-negative infections, particularly bloodstream infections, cause a high incidence of morbidity and mortality [1,2,3,4]. Of particular concern in these infections is the increasing development of resistance [5,6]. Several studies have evaluated the impact of gram negative bacteria on clinical patient outcomes, particularly in patients with elevated minimum inhibitory concentrations (MICs) to traditionally used antimicrobials [7,8,9,10,11,12].

According to the 2019 Centers for Disease Control and Prevention’s Antibiotic Resistance Threats in the United States, Enterobacteriaceae represent urgent (carbapenem-resistant) and serious (extended-spectrum beta-lactamase-producing) threats [13]. Chromosomally mediated AmpC-producing Enterobacteriaceae (CAE) are a group of bacteria that harbor inducible resistance to penicillins and first- through third-generation cephalosporins [14]. AmpC beta-lactamases are known to inactivate cefoxitin, as well as other cephalosporins, allowing clinicians an easy way to quickly identify a possible AmpC producer without running a genotypic test [15,16]. Cefoxitin resistance has a sensitivity and specificity of 97.4% and 78.7%, respectively, at identifying an AmpC-producing organism [16]. Therefore, many clinicians rely on this to provide insight as to the appropriate definitive coverage needed for these organisms. 

Relatively few studies have assessed the ideal treatment of these bacteria, and the majority focus on the usage of carbapenems or cefepime [17,18,19,20]. Extended-spectrum penicillin-β-lactamase inhibitor combinations have also been evaluated with conflicting evidence regarding effectiveness and appropriateness of these agents [21]. Although fluoroquinolones have been utilized for the treatment of CAE bloodstream infections, there is little information on the efficacy of these agents and no direct comparisons regarding differences in clinical outcomes between these agents and comparators [17,20,21]. The AmpC β-lactamase does not confer resistance to fluoroquinolones and some studies have reported an overall low incidence of fluoroquinolone resistance among these organisms [22,23]. The high oral bioavailability of fluoroquinolones makes them a reasonable option for oral stepdown therapy in certain patients despite no studies directly comparing treatment outcomes to other agents [24]. Therefore, our objective was to compare treatment outcomes between fluoroquinolone and nonfluoroquinolone definitive treatment of bloodstream infection caused by CAE.

## 2. Results

Ninety-eight patients had a blood culture positive for CAE during the study period, but 42 were excluded because they receive combination definitive therapy. A total of 56 patients were included in the study, where 31 (55%) patients received a fluoroquinolone (FQ) as definitive therapy, and 25 (45%) received definitive therapy with a non-FQ. All non-FQ definitive therapy patients received a beta-lactam (BL): 4 (16%) ceftriaxone, 1 (4%) ceftazidime, 9 (36%) cefepime, 3 (12%) piperacillin-tazobactam, 1 (4%) ertapenem, and 7 (28%) meropenem. The patients who received a FQ received either ciprofloxacin (*n* = 18; 58%) or levofloxacin (*n* = 13; 42%). The median age was 52 [IQR 41–62] years, and 32 (57%) patients were male. Over half of the population was African American (*n* = 29; 52%), and 25 (45%) patients were Caucasian. However, there were numerically more African Americans in the BL group (64% vs. 42%; *p* = 0.100), while there were numerically more Caucasians in the FQ group (55% vs. 32%; *p* = 0.087). The median Charlson Comorbidity Index was 2 [IQR 1–4], with no significant differences between groups. Patients in the FQ group had a significantly lower Pitt Bacteremia Score than those in the BL group (2 [IQR 1–3] vs. 4 [IQR 1–7]; *p* = 0.038). Other baseline characteristics were similar between groups (Table 1), with the exception that more patients with heart disease received a FQ than a BL for definitive therapy (36% vs. 8%; *p* = 0.015).

Over half of the infections originated within the hospital setting (*n* = 32; 57%) with significantly fewer patients in that group receiving FQ definitive therapy (45% vs. 72%; *p* = 0.044). Twenty-nine (52%) patients were admitted to the ICU (41.9% FQ vs. 64% BL, *p* = 0.100), and 14 (25%) received an infectious diseases consult (19.4% vs. 32%, respectively, *p* = 0.272). The maximum recorded infection-related laboratory markers were not significantly different between groups: temperature (102.3 °F FQ vs. 102.5 °F BL; *p* = 0.895), white blood cell count (12.8 cells/mm^3^ FQ vs. 11.9 cells/mm^3^ BL; *p* = 0.603), procalcitonin (12.1 ng/mL FQ vs. 24.8 ng/mL BL; *p* = 0.429). For most patients (30%), the presumed source of infection was unknown, however, 12 (21%) patients had a suspected central line infection, and 11 (20%) patients had a suspected urinary tract infection (Table 2). Adequate source control was achieved in 14/20 (70%) of applicable infections. *S. marcescens* was the predominant organism isolated (*n* = 27; 48%), followed by *Enterobacter* spp. (*n* = 26; 46%) and *M. morganii* (*n* = 3; 5%). *Citrobacter* spp. and *Providencia* spp. were not isolated in any included patients. The MICs were obtained and results are displayed in Table 3, with no differences found between groups. The median duration of bacteremia was 3 [IQR 2–4] days and was not different between groups (*p* = 0.385), however, 5 (9%) patients had a persistent bacteremia (*p* = 0.658). 

The predominant empiric therapy was piperacillin-tazobactam (41%), followed by meropenem (23%), cefepime (23%), and levofloxacin (21%). Empiric therapy was not significantly different between groups, except that significantly more FQ patients received a FQ empirically (32% vs. 8%; *p* = 0.028). Most patients (68%) received definitive therapy intravenously, and this occurred more often in the BL group than the FQ group (100% vs. 42%; *p* < 0.001). Clinically and numerically, patients in the FQ group had a longer duration of therapy than the BL group (12 days vs. 8 days; *p* = 0.483).

Clinical failure occurred in 10 (18%) patients, with 4 (13%) in the FQ group and 6 (24%) in the BL group (*p* = 0.315) (Table 4). There was no statistical difference in 90-day readmissions between groups (7% FQ vs. 17% BL; *p* = 0.387). One patient expired while admitted. Microbiological cure occurred in 55 (98%) patients, and only 1 (2%) patient experienced a recurrence. The median length of stay was 17 [IQR 7–37] days, with a significantly longer length of stay in the BL group (*p* = 0.001). Additionally, the median infection-related length of stay was 10 [IQR 6–20] days, and, again, a significantly longer stay occurring in the BL group (*p* = 0.002).

## 3. Discussion

Several studies have reviewed the treatment of CAE, but many have focused on treatment with carbapenems or other broad-spectrum beta-lactams. Although beta-lactams are generally preferred, there are situations where alternative options are warranted. To our knowledge, this is one of the first studies directly examining the utility of the fluoroquinolones versus nonfluoroquinolone (beta-lactam) therapy. 

In this study, fluoroquinolones had a numerically lower rate of composite clinical failure than comparators (12.9% vs. 24%), but this difference was not statistically significant. The rates of clinical failure in our groups were lower than a similar study by Derrick et al. (17.9% vs. 33.8%), which described overall treatment failure in patients with AmpC-producing Enterobacterales who received either third-generation cephalosporins or comparator therapy [25]. In addition, 90-day readmission rates were low (10.9% overall) but were numerically higher in the beta-lactam group (6.5% vs. 16.7%). In the study by Derrick et al., 90-day reinfection rates were comparable (9.2%). Finally, despite the fact that the fluoroquinolone group had a longer duration of therapy, both total (9 vs. 23 days) and infection-related (7 vs. 16) lengths of stay in this study were significantly shorter in the fluoroquinolone group versus comparators. It is likely that the numerical differences between groups in our study are due to differences in disease acuity. Charlson comorbidity index was similar between groups, but the Pitt Bacteremia Score was significantly higher in the beta-lactam group (2 vs. 4, respectively). This suggests that the beta-lactam group had more severe disease [26]. Additionally, despite over half of the population having a hospital-acquired bacteremia, empiric and definitive fluoroquinolone therapy has been shown to decrease the time to defervescence when compared to third-generation cephalosporins [27]. While we did not examine time to defervescence, many clinicians will utilize the presence of a fever as an indicator that the current therapy is not adequate, resulting in a broadening or change of coverage. 

Although a recent study by Henderson et al. demonstrated that a Pitt Bacteremia Score of ≥4 is predictive of mortality in nonbacteremia infections, the all-cause mortality in this study was low (1.8%) [26]. In two meta-analyses evaluating treatment of Enterobacteriaceae [17,28], mortality was higher than our study. In a meta-analysis focusing on treatment of *Enterobacter*, *Citrobacter*, or *Serratia* species by Harris et al., a lower risk of mortality was seen in patients receiving fluoroquinolones for definitive therapy (9% vs. 13.5% in a pooled analysis), but authors caution that this was likely due to less serious illness in this cohort [17]. In another meta-analysis evaluating Enterobacteriaceae producing extended-spectrum beta lactamases, mortality was similar between groups receiving fluoroquinolones or carbapenems for definitive therapy (16% vs. 13%) [28]. The low mortality seen in our study is likely due to the small sample size.

As with any study, ours is not without limitations. Due to a more restrictive definition of definitive therapy, we had a small sample size, which may limit external validity. However, this definition allowed us to specifically isolate the impact of fluoroquinolone monotherapy for Gram-negative bloodstream infections. Lastly, while a randomized clinical trial would be optimal, they are typically not feasible to conduct. Therefore, a retrospective, observational design was employed. This design could have led to missing data and incomplete records. 

To expand this study in the future, there are several possible considerations. First, the definition of “definitive therapy” could be broadened, or patients receiving combination definitive therapy could be included. Next, we used a fairly restrictive definition of CAE. Samples could be sent for sequencing to definitively identify AmpC-producing strains for inclusion for clinical evaluation. Additionally, the definition of clinical failure can vary from study to study, and outcomes could be different pending definition. Finally, this study could be expanded to incorporate patients from multiple centers in order to increase sample size and the ability to extrapolate these results externally. 

## 4. Materials and Methods

### 4.1. Study Design, Setting, Patient Population

This retrospective cohort study was conducted at the University of Mississippi Medical Center. The study protocol was approved by the local institutional review board (protocol number 2019-0029). Adult patients admitted between 1 June 2012 to 15 October 2018, with a blood culture positive for CAE (*Enterobacter* spp., *Citrobacter* spp. (except *C. koserii*), *Serratia marcescens*, *Morganella morganii*, or *Providencia stuartii*) and who received inpatient antibiotic treatment for ≥48 h were included in the study. Patients that had polymicrobial bacteremia, combination definitive therapy, pregnant patients, hospice or palliative care patients, and patients that expired within 48 h of the initial positive culture were excluded. The primary outcome was to evaluate the difference in rates of clinical failure between patients who definitively received fluoroquinolone versus nonfluoroquinolone treatment for a bloodstream infection caused by CAE. Secondary endpoints included time to microbiological cure, infection-related length of stay, 90-day readmission, all-cause inpatient mortality, development of resistance during therapy, and rates of adverse drug reactions.

### 4.2. Study Variables and Definitions

Data collected included patient demographics, microbiological characteristics, treatment strategies, clinical and microbiological response, presence of infectious diseases consultation, hospital length of stay, readmission and recurrence of bacteremia. Presumed source of infection was determined from culture results and progress notes. The following bacteria are known to produce chromosomal AmpC β-lactamase enzymes and were defined as CAE for purposes of this study: *Enterobacter* spp., *Citrobacter* spp. (except *C. koserii*), *Serratia marcescens*, *Morganella morganii*, or *Providencia stuartii*. This definition was based on previously defined phenotypic definition of AmpC [14,29,30]. A modified phenotypic definition of cefoxitin resistance was used, as our institution does not routinely test these organisms for susceptibility to ampicillin, ampicillin-sulbactam or cefazolin. Community-acquired infection was defined as an initial positive blood culture that was drawn within 48 h of admission, whereas hospital-acquired infection was defined as positive blood cultures drawn ≥48 h after admission. Healthcare-associated infection was defined as community-acquired infection plus the presence of a medical device in situ. Microbiological cure was defined as negative blood cultures following the initial positive culture. Clinical failure was defined as the composite of readmission after 90 days, recurrence of bacteremia (new culture positive for CAE that is separated by at least 7 days [but no more than 30 days] from the last positive blood culture for CAE with at least one negative blood culture in the interim period), in-patient infection-attributable mortality, and/or persistent bacteremia (continuously positive blood cultures for >72 h of definitive therapy). Empiric therapy was defined as therapy initiated prior to the availability of blood culture results. Definitive therapy was defined as the final antibiotic chosen to complete a treatment course after the availability of antimicrobial susceptibilities.

### 4.3. Statistical Analysis

The study endpoints were examined using descriptive and inferential statistics. Statistical analysis was performed using SPSS software version 24.0 (IBM). Categorical data were analyzed using Chi-Square or Fisher’s Exact test, and continuous data were analyzed using Student’s *t*-test or Mann–Whitney U test, as appropriate. An alpha of 0.05 was deemed statistically significant. Variables that have a *p*-value < 0.2 on univariate analysis or deemed clinically relevant by the investigators were evaluated for inclusion in a multivariable logistic regression model. A ratio of 10:1 was used to determine the maximum number of allowable variables in the model. 

## 5. Conclusions

These results suggest that fluoroquinolones do not adversely impact clinical outcomes in patients with CAE. When alternatives to beta-lactam therapy are needed, fluoroquinolones may provide an effective option.

## Figures and Tables

**Table 1 antibiotics-09-00331-t001:** Baseline Demographics of Patients Receiving Fluoroquinolone versus Beta-Lactam Definitive Therapy for Chromosomally mediated AmpC-producing Enterobacteriaceae (CAE) Bloodstream Infections.

Variable, *n*(%) or Median [IQR ^1^]	Total (*n* = 56)	Fluoroquinolone (*n* = 31)	Beta-Lactam (*n* = 25)	*p*-Value
Age, years	52 [41.25–62]	50 [41–60]	55 [42.5–62.5]	0.360
Sex, male	32 (57.1)	18 (58.1)	14 (56)	0.877
Race				
Caucasian	25 (44.6)	17 (54.8)	8(32)	0.087
African American	29 (51.8)	13 (41.9)	16 (64)	0.100
Asian	1 (1.8)	1 (3.2)	0(0)	1.000
Other	1(1.8)	0(0)	1 (4)	0.446
Weight	81.6 [74.93–97.65]	81.6 [75.3–96.6]	81.6 [73.25–98.05]	0.941
Charlson Comorbidity	2 [1–4]	2 [0–6]	2 [1.5–3]	0.874
Pitt Bacteremia Score	3 [1–5]	2 [1–3]	4 [1–7]	0.038
Antibiotic Allergy	41 (73.2)	21 (67.7)	20 (80)	0.303
Penicillins	11 (19.6)	8 (25.8)	3 (12)	0.312
Cephalosporins	4 (7.1)	4 (12.9)	0 (0)	0.120
Aminoglycosides	2 (3.6)	2 (6.5)	0 (0)	0.497
Trimethoprim-Sulfamethoxazole	4 (7.1)	2 (6.5)	2 (8)	1.000
Clindamycin	1 (1.8)	1 (3.2)	0 (0)	1.000
Vancomycin	2 (3.6)	2 (6.5)	0 (0)	0.497
Macrolides	2 (3.6)	2 (6.5)	0 (0)	0.497
Tetracyclines	1 (1.8)	1 (3.2)	0 (0)	1.000
Comorbidities				
Heart Disease	13 (23.2)	11 (35.5)	2 (8)	0.015
Chronic Kidney Disease	10 (17.9)	7 (22.6)	3 (12)	0.485
Diabetes Mellitus	15 (26.8)	10 (32.3)	5 (20)	0.303
Cirrhosis	1 (1.8)	1 (3.2)	0 (0)	1.000
Cancer	9 (16.1)	6 (19.4)	3 (12)	0.716
Immunocompromised	4 (7.1)	3 (9.7)	1 (4)	0.620
Central venous catheter	22 (39.3)	12 (38.7)	10 (40)	0.922
Urologic	3 (5.4)	2 (6.5)	1 (4)	1.000

^1^ IQR = interquartile range.

**Table 2 antibiotics-09-00331-t002:** Infection and Microbiological Characteristics of Patients Receiving Fluoroquinolone versus Beta-Lactam Definitive Therapy for CAE Bloodstream Infections.

Variable, *n* (%) or Median [IQR ^1^]	Total (*n* = 56)	Fluoroquinolone (*n* = 31)	Beta-Lactam (*n* = 25)	*p*-Value
Presumed Source of Infection				
Unknown	17 (30.4)	9 (29)	8 (32)	0.810
Urinary Tract	11 (19.6)	8 (25.8)	3 (13)	0.312
Lower Respiratory Tract	6 (10.7)	2 (6.5)	4 (16)	0.391
Skin and Soft Tissue	4 (7.1)	2 (6.5)	2 (8)	1.000
Central Line	12 (21.4)	8 (25.8)	4 (16)	0.374
Intra-Abdominal	3 (5.4)	2 (6.5)	1 (4)	1.000
Source Control				
Not applicable	36 (64.3)	21 (67.7)	15 (60)	0.548
Adequate	14 (25)	6 (19.4)	8 (32)	0.277
Inadequate	6 (10.7)	4 (12.9)	2 (8)	0.682
Development of Resistance	1 (1.8)	1 (3.2)	0 (0)	1.000
Multiple Positive Cultures	5 (9.1)	2 (6.7)	3 (12)	0.650
Duration of Bacteremia, days	3 [2–4]	3 [1–3]	2.5 [2–4.25]	0.385
Persistent Bacteremia	5 (9.4)	2 (7.1)	3 (12)	0.658

^1^ IQR = interquartile range.

**Table 3 antibiotics-09-00331-t003:** Minimum Inhibitory Concentrations (µg/mL) in CAE Bloodstream Infections.

Variable, Median [IQR ^1^]	Total (*n* = 56)	Fluoroquinolone (*n* = 31)	Beta-Lactam (*n* = 25)	*p*-Value
Amikacin	2 [2–2]	2 [2–2]	2 [2–2]	0.720
Ampicillin	*n* = 3 32 [32–32]	*n* = 2 32, 32	*n* = 1 32	1.000
Ampicillin-Sulbactam	*n* = 3 16	*n* = 2 16, 16	*n* = 1 16	1.000
Cefepime	1 [1–1]	1 [1–1]	1 [1–1]	0.434
Cefoxitin	24 [16–64]	16 [16–64]	32 [16–64]	0.352
Ceftazidime	1 [1–1]	1 [1–1]	1 [1–1]	0.397
Ceftriaxone	1 [1–1]	1 [1–1]	1 [1–1]	0.684
Ciprofloxacin	0.25 [0.25–0.25]	0.25 [0.25–0.25]	0.25 [0.25–0.25]	0.907
Gentamicin	1 [1–1]	1 [1–1]	1 [1–1]	1.000
Levofloxacin	0.12 [0.12–0.12]	0.12 [0.12–0.12]	0.12 [0.12–0.12]	0.745
Meropenem	0.25 [0.25–0.25]	0.25 [0.25–0.25]	0.25 [0.25–0.25]	0.095
Tobramycin	1 [1–2]	1 [1–2]	1 [1–1.5]	0.345
Piperacillin-Tazobactam	*n* = 10 4 [4–25]	*n* = 5 4 [4–66]	*n* = 5 4 [4–66]	1.000
Trimethoprim-Sulfamethoxazole	20 [20–20]	20 [20–20]	20 [20–20]	0.434

^1^ IQR = interquartile range

**Table 4 antibiotics-09-00331-t004:** Clinical Outcomes of Patients Receiving Fluoroquinolone versus Beta-Lactam Definitive Therapy for CAE Bloodstream Infections.

Variable, *n* (%) or Median [IQR ^1^]	Total (*n* = 56)	Fluoroquinolone (*n* = 31)	Beta-Lactam (*n* = 25)	OR (95% CI) *p*-Value
Primary Endpoint				
Clinical Failure–Composite	10 (17.9)	4 (12.9)	6 (24)	0.469 (0.116–1.892) 0.315
Secondary Endpoints				
90-Day Readmission	6 (10.9)	2 (6.5)	4 (16.7)	0.345 (0.058–2.066) 0.387
Time to First Readmission, days	26 [12.5–55.5]	14, 60	26 [10.25–49.25]	1.000
Multiple 90-day Readmissions	1/6 (16.7)	0/2 (0)	1/4 (25)	0.600 (0.293–1.227) 1.000
Inpatient Mortality (All-cause)	1 (1.8)	0 (0)	1 (4)	0.436 (0.323–0.589) 0.446
Time to Mortality, days	11	-	11	-
Microbiological Cure	55 (98.2)	30 (96.8)	25 (100)	1.833 (1.440–2.334) 1.000
Recurrence	1 (1.8)	1 (3.2)	0 (0)	0.545 (0.429–0.694) 1.000
Length of Stay, days				
Total	16.5 [7–37]	9 [6–24]	23 [13–48.5]	0.001
Infection-Related	9.5 [6–19.75]	7 [5–12]	16 [7–29]	0.002

^1^ IQR = interquartile range.

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
