# Peer review of "Fluoroquinolone Versus Nonfluoroquinolone Treatment of Bloodstream Infections Caused by Chromosomally Mediated AmpC-Producing Enterobacteriaceae"

_antibiotics, 2020, doi:10.3390/antibiotics9060331_

Round 1

Reviewer 1 Report

The manuscript by Sarah Grace Gunter et al., discusses the effect of Fluoroquinolone versus non-fluoroquinolone 2 treatment in BSI. Overall the study is fine and has the potential to be accepted. I have certain concerns that require attention:

  1. The primary concern is the novelty of the study. The authors need to highlight the importance of this study and tell what additional knowledge will be added with this study. It needs to be mentioned that what was previously unknown and how this study is different from previously published literature.
  2. Results do not clarify the symptoms or disease, on the basis of which patients were recruited.
  3. The number of patients is very low.
  4. Please specify the time span and place(s) of recruitment.
  5. In results, define "Pitt Bacteremia Score"
  6. Please discuss the methods of detection for specific bacterial infections.  

Author Response

Reviewer 1:

The manuscript by Sarah Grace Gunter et al., discusses the effect of Fluoroquinolone versus non-fluoroquinolone 2 treatment in BSI. Overall the study is fine and has the potential to be accepted.

Thank you for the comments. We have addressed your concerns below.

I have certain concerns that require attention:

  1. The primary concern is the novelty of the study. The authors need to highlight the importance of this study and tell what additional knowledge will be added with this study. It needs to be mentioned that what was previously unknown and how this study is different from previously published literature.

We have expanded the introduction to include this information.

  1. Results do not clarify the symptoms or disease, on the basis of which patients were recruited.

Patients were selected retrospectively based on positive blood cultures. The presumed source of infection reported in Table 2 explains where the bacteremia originated from.

  1. The number of patients is very low.

In order to assess the primary outcome, we only included organisms known to produce AmpC. In addition, we excluded patients with combination or multiple definitive therapies, which further limited our sample. However, we believe these two methodological decisions help control confounding variables and allow our results to be more accurately attributable to treatment effects. 

  1. Please specify the time span and place(s) of recruitment.

Patients who were admitted to the University of Mississippi Medical Center between June 1, 2012 and October 15, 2018 were included. This information is included in section 4.1. Study Design, Setting, Patient Population.

  1. In results, define "Pitt Bacteremia Score"

The Pitt Bacteremia Score is a validated scoring system that has been used for 3 decades to measure acute severity of illness and predict mortality in patients with bloodstream infections, and the authors disagree that a definition is required based on the following references supporting the use of the scoring system: Am J Med 1989;87:540–6. Surgery 1991; 109:62–8. Ann Intern Med

1991; 115:585–90. Antimicrob Agents Chemother 1992; 36:2639–44. Int J Antimicrob Agents

1999; 11:7–12.

  1. Please discuss the methods of detection for specific bacterial infections.  

We have added the following line to the section 4.2: Study Variables and Definitions:

Presumed source of infection was determined from culture results and progress notes.

Reviewer 2 Report

The manuscript entitled “Fluoroquinolone versus non-fluoroquinolone treatment of bloodstream infections caused by chromosomally-mediated AmpC-producing Enterobacteriaceae” describes the outcomes of using fluoroquinolones to treat AmpC-producing Enterobacteriaceae.  This manuscript is well written and adds interesting findings to the literature.  There are some issues that should be addressed however as follows:

  1. The first sentence of the abstract should be reworded. Bacteria display sensitivity or resistance to antibiotics, antibiotics do not display sensitivity or resistance to bacteria.  Please reword this sentence for clarity.
  2. It would be helpful to include the methodology by which antibiotic susceptibility testing (AST)/confirmation of isolates as AmpC-producing were performed.
  3. It would be valuable to add a table summarizing AST results.
  4. The introduction and discussion sections of this manuscript as well as the references should be expanded unless this manuscript is intended to be a short communication.
  5. While the sample size reported is small, the findings of this study are valuable. The authors should state how this study can be expanded to include more patients as well as their future directions with this research.

Author Response

Reviewer 2:

Comments and Suggestions for Authors

The manuscript entitled “Fluoroquinolone versus non-fluoroquinolone treatment of bloodstream infections caused by chromosomally-mediated AmpC-producing Enterobacteriaceae” describes the outcomes of using fluoroquinolones to treat AmpC-producing Enterobacteriaceae.  This manuscript is well written and adds interesting findings to the literature. 

Thank you for the comments. We have addressed your concerns below.

There are some issues that should be addressed however as follows:

  1. The first sentence of the abstract should be reworded. Bacteria display sensitivity or resistance to antibiotics, antibiotics do not display sensitivity or resistance to bacteria.  Please reword this sentence for clarity.

The first sentence of the abstract has been reworded as follows:

Chromosomally-mediated AmpC-producing Enterobacteriaceae (CAE) display high susceptibility to fluoroquinolones; minimal clinical data exist supporting comparative clinical outcomes.

  1. It would be helpful to include the methodology by which antibiotic susceptibility testing (AST)/confirmation of isolates as AmpC-producing were performed.

The following paragraph was added to the methodology section:

The following bacteria are known to produce chromosomal AmpC β-lactamase enzymes and were defined as CAE for purposes of this study: Enterobacter spp., Citrobacter spp. (except C. koserii), Serratia marcescens, Morganella morganii, or Providencia stuartii. This definition was based on previously defined phenotypic definition of AmpC [Jacoby 2009, Hammer 2016, Livermore 1998]. A modified phenotypic definition of cefoxitin resistance was used, as our institution does not routinely test these organisms for susceptibility to ampicillin, ampicillin-sulbactam or cefazolin.

  1. It would be valuable to add a table summarizing AST results.

Thank you for the suggestion. We’ve referenced the MICs beginning in line 95 and added Table 3. We’ve also renumbered and referenced Table 4 within the text.

  1. The introduction and discussion sections of this manuscript as well as the references should be expanded unless this manuscript is intended to be a short communication.

Thank you for the suggestion. We have expanded the introduction and discussion, plus added references.

  1. While the sample size reported is small, the findings of this study are valuable. The authors should state how this study can be expanded to include more patients as well as their future directions with this research.

Thank you for the suggestion. A new paragraph including this information has been added to the discussion.

Reviewer 3 Report

My main concern is the sample size, which is really small, so it would be difficult to extrapolate results to other hospital centers.

On the other hand, they do not explain how they reach such a low number of cases: there is no flow diagram to know which patients were excluded and why. Regarding clinical outcomes, odds ratios have not been calculated either.

The authors should come into contact with researchers from other hospitals with similar characteristics to collect more data and reinforce their conclusions. Table headings  should be better defined; they are very concise, and they do not give appropriate information.

Author Response

Reviewer 3:

  1. My main concern is the sample size, which is really small, so it would be difficult to extrapolate results to other hospital centers.

This information has been included in the limitations section.

  1. On the other hand, they do not explain how they reach such a low number of cases: there is no flow diagram to know which patients were excluded and why.

We have included this information at the beginning of the results section.

  1. Regarding clinical outcomes, odds ratios have not been calculated either.

Odds ratios (95% CI) have been added to all nominal clinical outcomes within Table 4.

  1. The authors should come into contact with researchers from other hospitals with similar characteristics to collect more data and reinforce their conclusions.

This is a great suggestion for future research. We have included it in a new paragraph in the discussion.

  1. Table headings should be better defined; they are very concise, and they do not give appropriate information.

Table headings have been expanded.

Round 2

Reviewer 3 Report

The unit mcg (heading table 3) should be corrected to µg. The manuscript has been improved and the authors have followed all the suggestions.